# Mapping climate adaptation corridors for biodiversity—A regional-scale case study in Central America

Ian M. McCullough[1,2☯]*, Christopher Beirne[1,2☯], Carolina Soto-Navarro[1,2,3], Andrew Whitworth[1,2,4,5]*

1 Osa Conservation, Washington, DC, United States of America, 2 Osa Conservation Campus, Puntarenas, Costa Rica, 3 UN Environment Programme World Conservation Monitoring Centre (UNEP-WCMC), Cambridge, United Kingdom, 4 Institute of Biodiversity, Animal Health and Comparative Medicine, College of Medical, Veterinary and Life Sciences, University of Glasgow, Glasgow, Scotland, United Kingdom, 5 Department of Biology, Center for Energy, Environment and Sustainability, Wake Forest University, Winston-Salem, NC, United States of America

☯ These authors contributed equally to this work.
* andywhitworth@osaconservation.org (AW); immccull@gmail.com (IMM)

**Data Availability Statement:** Data and R code used for this study are publicly available on Zenodo: McCullough, I., & Beirne, C. (2024).

## Abstract

Climate adaptation corridors are widely recognized as important for promoting biodiversity resilience under climate change. Central America is part of the Mesoamerican biodiversity hotspot, but there have been no regional-scale analyses of potential climate adaptation corridors in Central America. We identified 2375 potential corridors throughout Central America that link lowland protected areas ($\leq$ 500 m) with intact, high-elevation forests ($\geq$ 1500 m) that represent potential climate change refugia. Whereas we found potential corridors in all Central American countries, potential corridors in Panama, Belize, and Honduras were most protected (medians = 64%, 49%, and 47%, respectively) and potential corridors in El Salvador were least protected (median = 10%). We also developed a corridor priority index based on the ecological characteristics and protected status of potential corridors and their associated start and end points. Compared to low- and medium-priority corridors, high-priority corridors (n = 160; top 7% of all corridors) were generally more protected, forested, and distributed across wider elevational gradients and more Key Biodiversity Areas, but also generally linked larger lowland protected areas to target areas that were larger, more protected, and spanned wider elevational gradients. For example, based on median values, high-priority corridors were 9% more protected and overlapped with 2–3 more Key Biodiversity Areas than low- and medium-priority corridors. Although high-elevation targets spanned considerably wider elevational gradients than lowland protected areas (medians = 695 vs. 142 m, respectively) and thus may be more likely to support refugia, they were considerably smaller than lowland protected areas (medians = 11 vs. 50 km$^2$ respectively) and mostly unprotected (median = 4% protection). This initial, regional assessment can help prioritize locations for finer-scale research, conservation, and restoration activities in support of climate adaptation corridors throughout Central America and highlights the need for greater conservation of potential high-elevation refugia.

Central America climate adaptation corridor analysis [Data set]. In Mapping climate adaptation corridors for biodiversity - A regional-scale case study in Central America (1.0). Zenodo. https://doi.org/10.5281/zenodo.11150568.

**Funding:** This research was supported by the International Conservation Fund of Canada, the Bobolink Foundation, the BAND Foundation, and the Gordon and Betty Moore Foundation. BAND Foundation: https://bandfdn.org/ Bobolink Foundation: https://www.bobolinkfoundation.org/ International Conservation Fund of Canada: grant #2021-63; https://icfcanada.org/ Gordon and Betty Moore Foundation: grant #9075; https://www.moore.org/ All of these were gifts to Friends of the Osa (Osa Conservation) rather than to individual authors of this study. The BAND Foundation and Bobolink Foundation are small, family-run organizations that do not issue grant numbers per se. Sponsors or funders played no role in the study design, data collection and analysis, decision to publish, or preparation of the manuscript.

**Competing interests:** The authors have declared that no competing interests exist.

## Introduction

Biological corridors refer broadly to landscape features that facilitate habitat connectivity for biodiversity, particularly across fragmented landscapes often used for agriculture [1]. Growing scientific evidence demonstrates the important role of biological corridors in climate adaptation by facilitating species range shifts [2–5]. Hence, an emerging conservation strategy is to create "climate adaptation corridors" that allow species to access suitable habitat at mid to high latitudes and elevations, often by linking protected areas [6–8]. The basis of this strategy has arisen from a body of science related to paleogeography, paleoecology, and past, present, and future climate and species distribution modeling [9–11]). Mountainous, complex landscapes are often the subject of climate adaptation corridor research owing to their environmental heterogeneity, topographic and vegetative buffering capacity, and relative climatic stability that together can support ecologically diverse climate change refugia that allow species to survive and adapt under changing climatic conditions [12–15].

Globally, however, protected area connectivity is generally lacking [16, 17] and over 62% of tropical forests are already incapable of facilitating range shifts to analogous future climates [18]. Whilst there has been an increase to 7.84% of global protected area connectivity [19], this still falls short of Target 3 of the Kunming-Montreal Global Biodiversity Framework to protect and connect 30% of terrestrial, inland water, marine, and coastal areas by 2030 [20]. Therefore, considerable improvements in protected area connectivity from local to regional scales, likely via help from climate adaptation corridors, are necessary to promote biodiversity persistence under climate change.

Whereas global and pan-tropical assessments are useful for international policymaking and tracking of global conservation targets, such assessments are often too coarse to inform conservation policy and decision making at local to national scales. This is particularly the case for smaller countries, which may be highly biodiverse but easily overlooked in global assessments and policy forums. For example, Central America consists of several relatively small countries located within the Mesoamerican biodiversity hotspot, but has a history of deforestation (often for agriculture) and varying conservation policies among countries [21]. Currently, no Central American countries meet the 30% by 2030 protected area coverage and connectivity target, particularly in terms of connectivity [22] (S1 Table in S1 Appendix), even though connectivity has been a major conservation priority in previous decades.

Central American countries recognized the importance of connectivity for biodiversity with the adoption of the Mesoamerican Biological Corridor (MBC) in 1997, which originally sought to connect habitats from Mexico to Panama using the jaguar (*Panthera onca*), the region's largest carnivore, as a flagship and umbrella species as the basis for this approach [23]. Although the MBC was initially lauded for its ambitious international framework and successfully raised hundreds of millions of dollars, the envisioned regional corridor network was never truly realized due to a combination of factors including lack of coordination among countries, unclear guidelines for what constitutes a corridor, inconsistent or limited corridor monitoring criteria and data, insufficient buy-in among diverse stakeholders, and shifting priorities toward pursuing co-benefits between biodiversity and sustainable development [23–25]. Consequently, individual countries have largely operated on their own terms outside the original, regional framework with variable results. In the Mexican states of Chiapas and Tabasco, corridors were initially designed to become part of the MBC using ecological criteria, but ultimately were implemented in locations acceptable to local landowners and their effectiveness at facilitating ecological connectivity remains unknown [26]. El Salvador developed a National Biological Corridor management strategy, but corridors have largely remained an ambiguous concept and have rarely translated from planning to practice [25]. Otherwise, there

are many recent examples of research evaluating proposed corridors using ecological criteria in specific portions of Central America [27–29] but it is unknown to what extent such research might influence regional conservation planning or policymaking. What limited published research on regional or international connectivity in Central America has primarily focused on jaguars [30–33].

Costa Rica, whose National Biological Corridor Program consists of 44 corridors that cover 38% of the country, may be considered the most successful example of corridor implementation in Central America. Greater forest protection and restoration associated with corridors have improved mammal species richness and functional connectivity within the region [34]. Although forest fragmentation significantly decreased in Costa Rican corridors since their adoption, these corridors still mostly consist of unprotected, privately owned land too small and fragmented to support populations of key mammal species such as jaguars and white-lipped peccary (*Tayassu pecari*) [35]. Perhaps even more concerning given ongoing climate change is that these existing corridors, even within a country known for its forward-thinking approach to conservation, were not designed to encompass the wide elevational gradients necessary to facilitate species range shifts under climate change [35, 36]. Hence, coordinated, regional-scale strategies are needed to facilitate connectivity among protected and well conserved areas across elevational gradients and identify focal locations for local- to landscape-scale conservation efforts. This is particularly the case for small countries with limited protected areas or land available for additional protection [24].

In this study, we map the spatial distribution of potential climate adaptation corridors throughout Central America, assess their conservation status and ecological characteristics (e.g., length, elevational breadth, forest condition), and calculate a priority index to support climate adaptation corridor implementation efforts and identify promising areas for future, finer-scale research or conservation activities. We hypothesize that potential climate adaptation corridors exist throughout the region, but vary widely in terms of protection and ecological condition, necessitating identification of those with the greatest potential to facilitate regional biodiversity persistence under climate change. Our study can help bridge the gap between local- and global-scale studies and move toward realizing the original vision of regional-scale connectivity under the MBC by facilitating systematic prioritization among numerous potential climate adaptation corridors and providing important baseline data (e.g., forest condition) that could be used to monitor future outcomes of corridors throughout the region. Application of our findings can help build a climate adaptation corridor network across elevational gradients in Central America and guide conservation investments at multiple spatial scales.

## Materials and methods

### Study area

Our main study area consisted of 7 Central American countries: Belize, Costa Rica, El Salvador, Guatemala, Honduras, Nicaragua, and Panama (Fig 1). These countries encompass a total of 516,343 km$^2$, approximately 30% of which is protected [37]. We buffered this area by 500 km to allow corridors originating in our 7 focal countries to cross or terminate in southeastern Mexico or northwestern Colombia. These additional areas increased our overall study area to 1,032,074 km$^2$, approximately 21% of which is protected [37]. We did not consider oceanic or inland islands in our analysis.

### Overall workflow

First, we identified start and end nodes. Start nodes were lowland protected areas and end nodes were contiguous, relatively intact patches of highland forest (either protected or

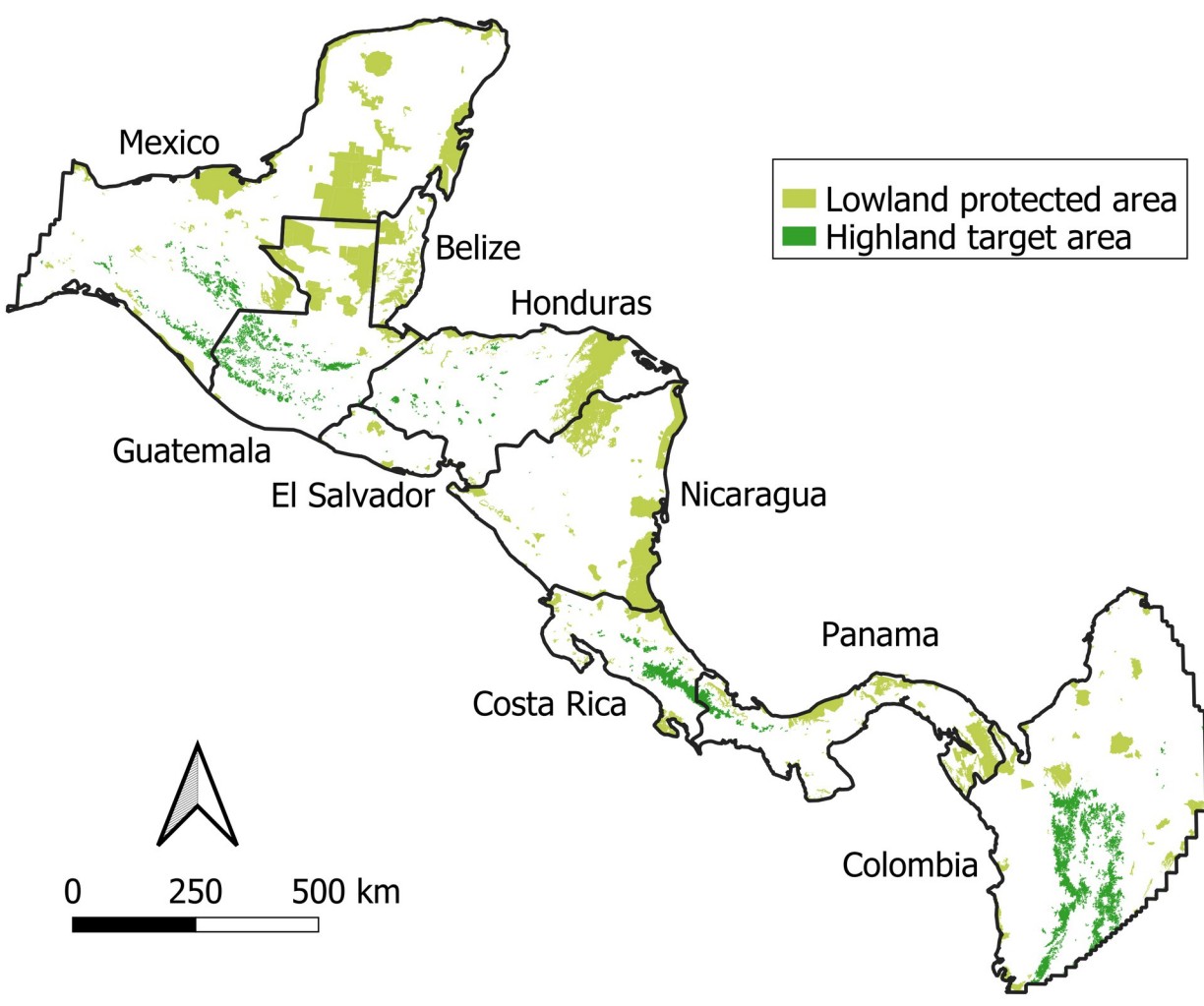

**Fig 1. Study area, lowland protected areas (starting nodes), and highland target areas (end nodes).** Country boundaries are public domain and were obtained from Natural Earth (50m-admin-0-countries-2) before clipping to our study area (https://www.naturalearthdata.com/). Lowland protected areas are polygons $\geq 5$ km$^2$ and $\leq 500$ m elevation from the World Database on Protected Areas [37]. Highland target areas are patches ($\geq 5$ km$^2$) of upper-montane ($\geq 1500$ m elevation) contiguous forest with medium or high Forest Landscape Integrity Index values [47].

unprotected) that represent potential climate change refugia. Second, we created a conductance surface for modeling least cost paths and third, we quantified the ecological characteristics and conservation status of potential climate adaptation corridors and their associated start and end nodes to develop a corridor priority index. We used Google Earth Engine [38] through the 'rgee' R package [39] to download data layers for connectivity assessments. Throughout our analysis, occasionally start and end nodes and often corridors spanned multiple countries, so we referred to these by their primary overlapping country when necessary to facilitate comparisons among countries. A permanent repository containing data and R code is available on Zenodo [40]. We used R version 4.3.0 for all analyses [41].

## Start and end nodes

We used the World Database on Protected Areas (WDPA) as the most up-to-date, comprehensive source of protected areas globally [37]. We used the 'wdpar' R package [42] to filter out

duplicated polygons and remove protected areas with unreported area, "proposed" status, or 'UNESCO-MAB Biosphere Reserve' designation. We included terrestrial protected areas and land portions of marine or "partial" protected areas. We also added the 'Belize Maya' protected area, a recently created, large protected area (958 km$^2$) not found in the WDPA at the time of access. We used the Shuttle Radar Topography Mission digital elevation data v4 [43] to identify protected areas below 500 m (mean), a threshold to define lowland forested habitats [44, 45]. We defined a single location as the start node within each lowland protected area using the 'st_point_on_surface' function in the 'sf' R package [46]. Because we operated at the regional scale, we only considered start nodes $\geq$ 5 km$^2$. Start nodes per country ranged from 14 (El Salvador) to 110 (Mexico) (S2 Table in S1 Appendix). In total, our analysis included 475 start nodes ranging from 5–7249 km$^2$ (median = 51 km$^2$) (S2 Table in S1 Appendix, Figs 1 and 2A).

We defined end nodes (target habitats) as patches ($\geq$ 5 km$^2$) of upper-montane ($\geq$ 1500 m) (mean) contiguous forest [44], with medium or high Forest Landscape Integrity Index values that represent relatively intact, potential highland climate change refugia [47]. This resulted in 529 end nodes ranging from 5–5938 km$^2$ (median = 10 km$^2$). End nodes were mostly located in Guatemala (183), Colombia (151), and Mexico (105), whereas Belize was the only country without end nodes (S2 Table in S1 Appendix). Similar to start nodes, we used the 'st_point_on_surface' function [46] to define end nodes within each target habitat. Unlike start nodes, however, we included both protected and unprotected end nodes because any contiguous, high-elevation forest may represent potential climate change refugia or candidate areas for future conservation. To prevent least cost paths needing to travel further to end nodes within larger target habitats, we seeded additional end nodes using hexagonal sampling in the 'st_sample' function of the 'sf' R package [46]. This resulted in a suite of 779 end points across the 529 main end node polygons with 1–41 end points per polygon (median = 1 point).

## Conductance surface and least cost path modeling

We developed a structural conductance surface (i.e., inverse of resistance) based on the European Space Agency's 10 m WorldCover 2020 v100 dataset [48]. We resampled the land cover data to 100 m resolution to optimize computational constraints. We assigned habitat-specific conductance values (S3 Table in S1 Appendix) based upon previous research conducted for three keystone, medium-large terrestrial mammals of conservation concern present across the study area: White-lipped peccary (crucial seed disperser and browser) [29]; Baird's tapir (crucial seed disperser and browser) [49]; and jaguar (apex predator) [30]. As the WorldCover 'forest' category encompasses a broad array of forest qualities and disturbances, we modified its conductance value by its standing biomass (derived from [50]), to penalize low quality (low biomass) habitats. To create the modifier, we thresholded all forest biomass values > 100 Mt C to 100 and divided biomass values by 100 such that 0 represented locations with zero biomass and 1 represented locations with $\geq$ 100 Mt C, then multiplied the conductance score by this modifier (S1 Fig in S1 Appendix). Like many previous regional- to continental-scale climate connectivity studies, we did not adjust conductance values for topography as our focus was long-term range shifts rather than short-term movements (e.g., [51–53]). Moreover, steeper corridors have greater capacity to help species overcome rapid climate change velocities, topography can influence individual species' movements positively or negatively [54], and our study lacks a single focal species on which to base terrain-derived conductance corrections. Similarly, we did not adjust conductance values for climate dissimilarity as previous "climate corridor" studies have done (e.g., [55]). Whereas "climate corridors" typically are designed to minimize exposure to climatically unsuitable habitat along routes for long-term range shifts,

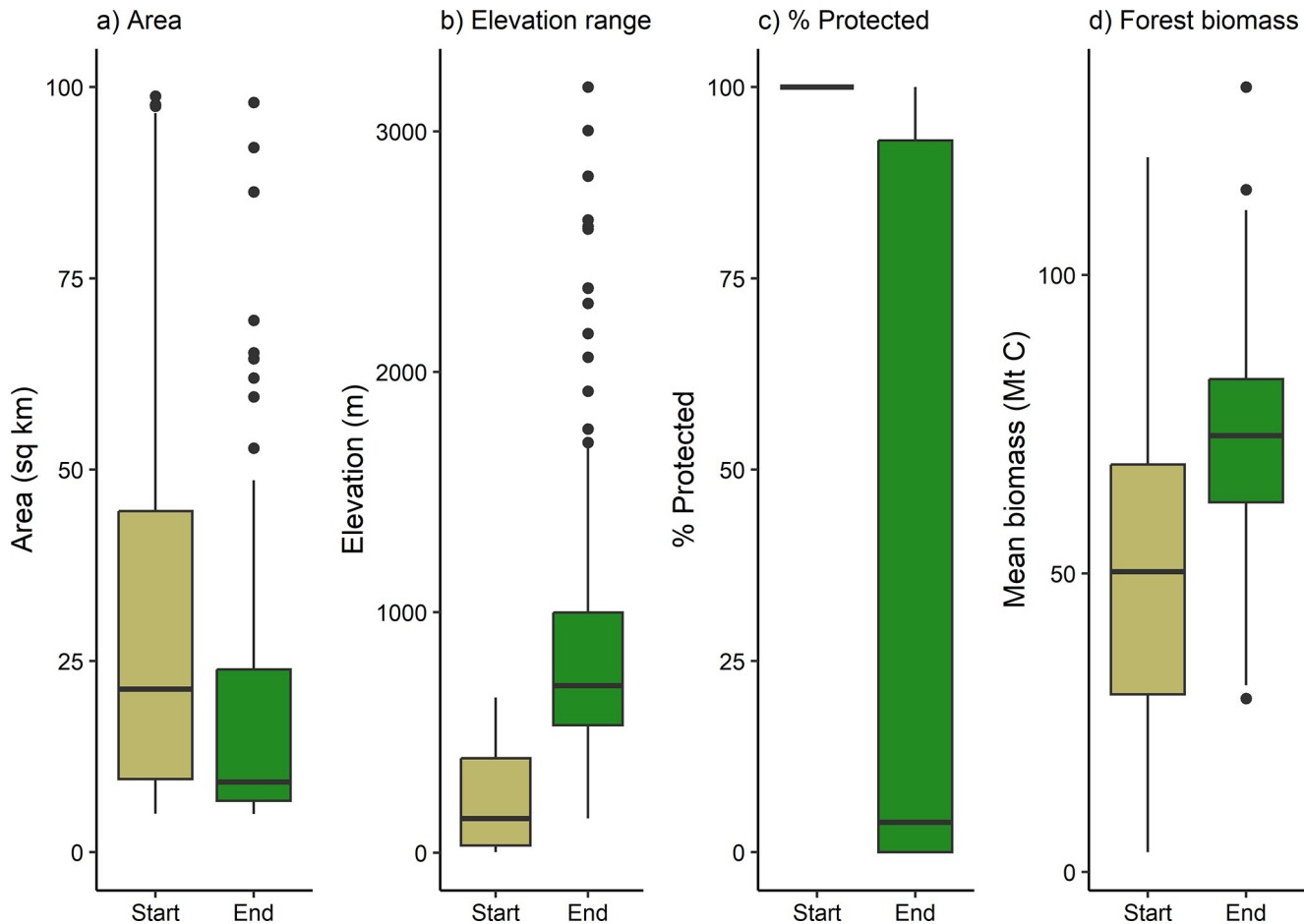

**Fig 2. Comparison of area, elevational range, protection, and mean forest biomass between start and end nodes.** Figure encompasses all start nodes (n = 475) and only the end nodes in which potential climate adaptation corridors terminated (n = 192).

the species on which our conductance surface is based already live across lowland and highland habitats in our region (i.e., wide temperature gradients).

We next applied least cost path modeling based on our conductance surface using the 'shortestPath' function in the 'gdistance' R package [56]. Whereas many landscape connectivity studies map least cost paths between each pair of start and end nodes, this was computationally impractical given our large spatial extent and hundreds of nodes. Therefore, from each of the 475 start nodes, we identified the 5 nearest end nodes (based on Euclidean distance) using the 'st_nn' R function [46] and mapped least cost paths for each of these 5 node pairs, resulting in a total of 2375 unique least cost paths.

## Ecological characteristics, conservation status, and prioritization of potential climate adaptation corridors

Whereas least cost paths inherently represent the path of least resistance from one location to another, individual pathways vary in their relative capacity to function as climate adaptation corridors. Therefore, we quantified several ecological characteristics and the conservation status of potential corridors to facilitate comparison among and prioritization of different potential corridors (S4 Table in S1 Appendix). Corridor variables included length, elevational range,

percentage of protection, number of overlapping protected areas (excluding marine) and Key Biodiversity Areas (KBAs), and mean forest biomass. We used a 1000 m buffer around corridor polylines to calculate all these variables (except corridor length) using the 'extract' function in the 'terra' R package [57] and vector operations (Buffer, Overlap analysis, and Intersection) in QGIS Desktop v3.24.1. We chose a 1000 m buffer (2 km corridor width) because this is considered a "rule-of-thumb" minimum corridor width for facilitating long-term gene flow and habitat recolonization [58]. Because variable distributions were often not normal, we used Spearman's correlation tests to compare different ecological and conservation variables.

Our climate adaptation corridor priority index was based on the premise that shorter, protected pathways with intact vegetation that link large lowland protected areas to large, high-elevation patches of contiguous forest offer the greatest potential as climate adaptation corridors, particularly if they overlap with areas of high biodiversity (e.g., KBAs). We therefore applied a combination of corridor, start node, and end node variables that reflected this premise (S4 Table in S1 Appendix) into a principal component analysis (PCA). We eliminated overlapping number of protected areas for each corridor and end node area because these were highly correlated with corridor length and end node elevational range, respectively (Spearman's rho = 0.72 and 0.76, respectively). We inverted corridor length such that shorter corridors would be prioritized and we normalized all input variables (mean of 0 and standard deviation of 1) prior to PCA calculations. Based on visual inspection of the screeplot, we decided to use the first 3 principal components, which collectively explained 63% of variation in the data, to calculate priority index values for each potential corridor. Index values were calculated as the distance from the origin within the 3-dimensional space defined by the first 3 principal components. We designated high-, medium-, and low-priority index values based on visual inspection of their distribution (see results) and the need to identify a relatively small number of high-priority potential climate adaptation corridors for future, finer-scale investigation. Finally, we also compared the ecological characteristics and conservation status of corridors and associated start and end nodes across high-, medium-, and low-priority corridor groups to aid interpretation. We applied nonparametric Kruskal-Wallis tests to compare differences among these groups and if there were significant differences, we then applied Dunn's test with Holm's p-value adjustment to perform pairwise comparisons using the 'dunn_test' function in the 'rstatix' R package [59].

# Results

## Basic characteristics of start and end nodes and potential climate adaptation corridors

Overall, end nodes were generally smaller, spanned wider elevational gradients, and had more intact forest biomass than start nodes, but were also poorly protected. Median start and end node sizes were 50 and 11 km$^2$, respectively (Fig 2A). Despite these notable size differences, end nodes spanned a median of 695 m in elevation, whereas start nodes only spanned a median of 142 m (Fig 2B). Because we selected start nodes as protected areas, these were all fully protected, but median end node protection was only 4% (Fig 2C), even though end nodes were selected as contiguous, intact forest patches (median = 73 Mt C) (Fig 2D). This result indicates that intact highland forest habitat is often unprotected in Central America. However, start node forest biomass was lower (median = 50 Mt C) despite greater protection. Finally, potential climate corridors originating from all 475 start nodes only terminated in 192 of the possible 529 end nodes (36%). This was because some pathways converged on the same end nodes and because we only modeled least cost paths between each start node and the 5 geographically closest end nodes.

Table 1. Summary statistics for potential climate adaptation corridors by country*.

| Country | Corridors | Length (km) | Elevation range (m) | Protection (%) | Overlapping protected areas | Overlapping KBAs | Countries crossed | Forest biomass (Mt C) |
|---|---|---|---|---|---|---|---|---|
| Belize | 194 | 185, 358, 860 | 1674, 2183, 2480 | 30, 49, 66 | 9, 19, 27 | 4, 6, 11 | 2, 3, 4 | 56, 65, 72 |
| Colombia^ | 317 | 14, 134, 805 | 1241, 1797, 3553 | 1, 19, 100 | 1, 3, 15 | 0, 1, 5 | 1, 1, 2 | 30 67, 117 |
| Costa Rica | 317 | 11, 81, 291 | 1400, 1978, 3440 | 4, 36, 100 | 1, 4, 11 | 1, 3, 7 | 1, 1, 2 | 34, 65, 97 |
| El Salvador | 64 | 30, 104, 200 | 1542, 1922, 2343 | 2, 10, 44 | 1, 4, 19 | 0, 2, 5 | 1, 2, 3 | 26, 39, 68 |
| Guatemala | 340 | 17, 159, 451 | 1527, 2020, 3942 | 1, 36, 100 | 1, 7, 20 | 0, 3, 8 | 1, 2, 3 | 17, 60, 83 |
| Honduras | 251 | 6, 141, 532 | 1310, 2033, 2685 | 6, 47, 100 | 1, 4, 24 | 0, 2, 10 | 1, 1, 3 | 24, 58, 81 |
| Mexico^ | 485 | 21, 276, 1158 | 1279, 1866, 2528 | 1, 35, 100 | 1, 5, 28 | 0, 3, 13 | 1, 2, 4 | 26, 58, 79 |
| Nicaragua | 145 | 104, 247, 510 | 1249, 1736, 2696 | 5, 29, 86 | 2, 6, 13 | 0, 3, 8 | 1, 2, 2 | 31, 52, 83 |
| Panama | 262 | 15, 172, 401 | 1296, 1720, 3392 | 5, 64, 100 | 1, 5, 14 | 1, 3, 9 | 1, 1, 2 | 33, 74, 100 |

* in columns 3–9, first, second, and third numbers are minimum, median and maximum values, respectively

^ Values are not for the entire country; our study area only included portions of southeastern Mexico and northwestern Colombia

We identified 2375 potential climate adaptation corridors, 1573 of which occurred primarily within our 7 focal countries (Table 1). The total number of corridors per country ranged from 64 in El Salvador to 485 in Mexico, despite only considering a fraction of Mexico in our analysis. Median corridor length ranged from 81 km (Costa Rica) to 358 km (Belize) and was 165 km across all countries (Table 1 and Fig 3A). The shortest corridor originating in each country ranged from 6 km (within Cerro Azul National Park, Honduras) to 185 km (Deep River Forest Reserve, Belize to Cusuco National Park, Honduras, crossing through Guatemala). Median corridor elevational breadth ranged from 1720 m (Panama) to 2183 m (Belize) and was 1867 m across all countries (Table 1 and Fig 3B). A total of 1312 corridors (55%) were within a single country, whereas 784 (33%), 197 (8%), and 82 (3%) corridors spanned 2, 3, and 4 countries, respectively. Corridors in Belize typically spanned the most countries (median = 3 countries), whereas in contrast, corridors in Colombia, Costa Rica, Honduras, and Panama typically were the most self-contained (median = 1 country) (Table 1). Overall, potential climate adaptation corridors were distributed throughout the region and varied widely in terms of length, elevational breadth, and countries spanned.

## Climate adaptation corridor protection and ecological characteristics

The protection and ecological characteristics of potential climate adaptation corridors were highly variable across the region. Corridors in Panama were most protected (median = 64%), followed by Belize (median = 49%), and Honduras (median = 47%) (Table 1 and Fig 3C). Corridors located in El Salvador were the least protected (median = 10%). Of the 36 fully protected corridors, Panama and Honduras had the most fully protected corridors (16 and 8, respectively), whereas Colombia, Costa Rica, and Mexico all had 3–5 fully protected corridors and 5 were shared between Panama and Colombia. Corridors located in Belize overlapped with the most protected areas (median = 19), with medians of 3–7 overlapping protected areas per corridor across all other countries (Table 1 and Fig 3D). Similarly, corridors located in Belize also overlapped with the most KBAs (median = 6), with medians of 2–3 overlapping KBAs for all other countries (except partial Colombia with a median of 1 KBA) (Table 1 and Fig 3E). Corridors in Belize were generally the longest (median = 358 km) (Fig 3A), which may explain the larger number of overlapping protected areas and KBAs. Both number of overlapping

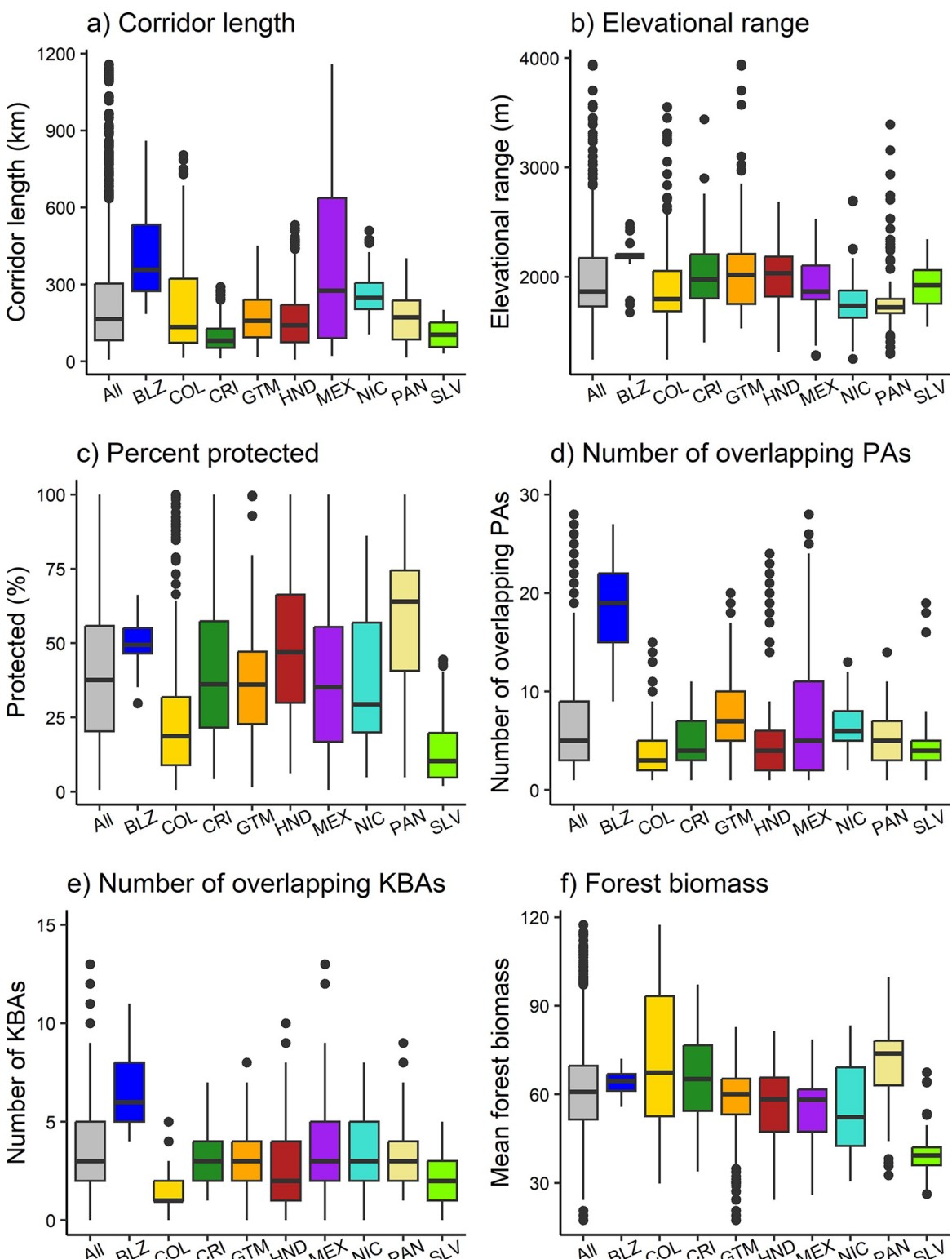

**Fig 3. Ecological characteristics and conservation status of potential climate adaptation corridors by country.** Note that our study area only included southeastern Mexico and northwestern Colombia. PA = protected area, KBA = Key Biodiversity Area, BLZ = Belize, COL = Colombia, CRI = Costa Rica, GTM = Guatemala, HND = Honduras, MEX = Mexico, NIC = Nicaragua, PAN = Panama, SLV = El Salvador.

protected areas and KBAs were positively correlated with corridor length across all countries (respective Spearman's rho = 0.76 and 0.75; p < 2.2e-16), but percentage corridor protection was not (Spearman's rho = -0.02, p = 0.32). Mirroring corridor protection, corridors in Panama and El Salvador had the greatest and least forest biomass, respectively (medians = 74 and 39 Mt C) (Table 1, Fig 3F). Across other countries, median corridor forest biomass ranged from 52 to 67 Mt C and percentage corridor protection was positively correlated with corridor forest biomass (Spearman's rho = 0.50, p < 2.2e-16). Corridors in Colombia and Costa Rica were among the most heavily forested (medians = 67 and 65 Mt C, respectively), behind only Panama, despite poor and moderate levels of corridor protection (Colombia: median = 19% protection; Costa Rica: median = 36% protection). In summary, corridors located in Panama, Belize, and Honduras were generally the most protected and those located in Panama, Colombia, Costa Rica, and Belize were generally the most forested. Longer corridors generally overlapped with the most protected areas and KBAs but were not necessarily more protected overall.

### Climate adaptation corridor priority index

Climate adaptation corridor priority index values ranged from 0.21–7.11 and followed a right-skewed distribution (S2 Fig in S1 Appendix). Considering this and that only a small percentage of the 2375 potential corridors we identified might be pursued for conservation efforts, we assigned priority index values < 2 as low (n = 1290; 54%), 2–4 as medium (n = 925; 39%), and > 4 as high (n = 160; 7%). We encountered several clusters of high-priority corridors, including the Yucatan Peninsula (mostly Mexico and Belize), the coastal lowlands to Sierra Madre in Guatemala, the coastal lowlands to the Cordillera Occidental of Colombia, eastern Honduras/northern Nicaragua, southeastern Nicaragua/northern Costa Rica, and converging on La Amistad International Park of Costa Rica and Panama from both the Pacific and Caribbean coasts (Fig 4). All countries except El Salvador contained at least one high-priority corridor, but El Salvador had a relatively high density of medium-priority corridors. Much of Nicaragua was covered by low-priority corridors, particularly the western portion of the country where there are few large protected areas.

Analysis of the ecological characteristics and conservation status of potential climate adaptation corridors according to priority index values demonstrated that high-priority corridors were significantly longer, spanned wider elevational gradients, had more forest biomass, were more protected, and overlapped with more protected areas and KBAs compared to medium- and low-priority corridors (Dunn's p ≤ 0.003) (Table 2 and Fig 5A–5F). For example, high-priority corridors spanned median elevational ranges of 2185 m (compared to 1868 m and 1829 m for medium- and low-priority corridors, respectively) (Table 2 and Fig 5B) and had median protection of 46% (compared to 37% and 38% for medium- and low-priority corridors, respectively) (Table 2 and Fig 5D). These findings were largely consistent with our prioritization design, except for corridor length, which was positively correlated with the number of overlapping protected areas and KBAs (Spearman's rho = 0.72 and 0.66, respectively; p < 2.2e-16). Additionally, high-priority corridors consistently led to end nodes that were larger, spanned wider elevational ranges, and were more protected compared to medium- and low-priority corridors (Table 2 and Fig 5G–5I) (Dunn's p ≤ 3.70e-15). Similarly, high-priority corridors consistently originated from larger start nodes compared to medium- and low-priority corridors (Dunn's p ≤ 1.23e-6), but start nodes of high-priority corridors also spanned narrower elevational ranges compared to those of medium- and low-priority corridors (Dunn's p ≤ 0.02) (Table 2, Fig 5J and 5K). However, all lowland protected areas have considerably shorter elevational gradients than highland target areas (Table 1 and Fig 2B). In summary,

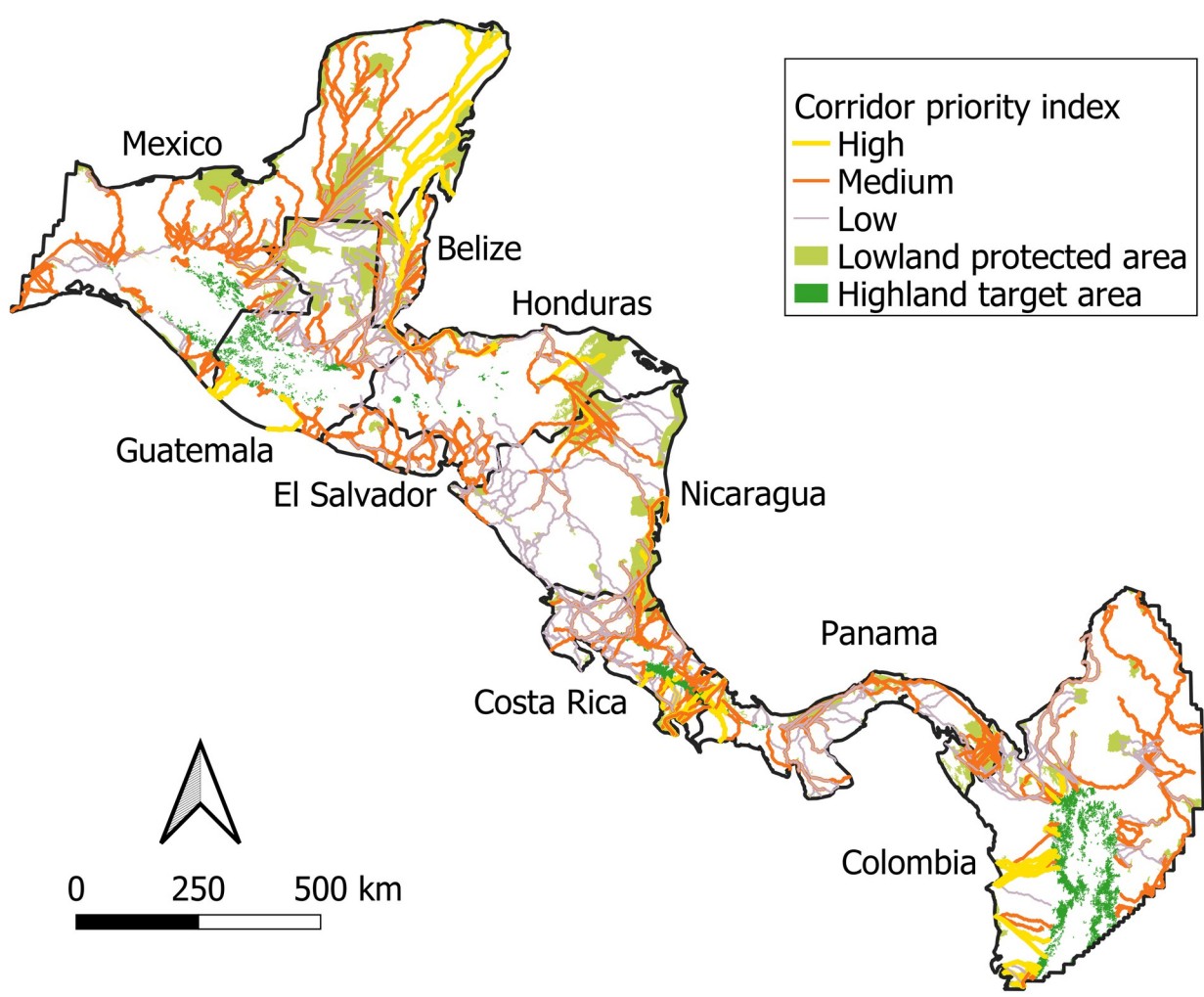

**Fig 4. Climate adaptation corridor priority index values.** Country boundaries are public domain and were obtained from Natural Earth (50m-admin-0-countries-2) before clipping to our study area (https://www.naturalearthdata.com/). Lowland protected areas are polygons $\geq$ 5 km$^2$ and $\leq$ 500 m elevation from the World Database on Protected Areas [37]. Highland target areas are patches ($\geq$ 5 km$^2$) of upper-montane ($\geq$ 1500 m elevation) contiguous forest with medium or high Forest Landscape Integrity Index values [47].

high-priority corridors were generally more protected, forested, and distributed across wider elevational gradients and more KBAs, and they also generally linked larger lowland protected areas to end nodes that were larger, more protected, and more variable in terms of elevation.

## Discussion

### Potential for climate change refugia in neotropical mountainous landscapes

This study represents one of the few regional-scale connectivity analyses in Central America and is, to the best of our knowledge, the first to prioritize potential climate adaptation corridors across elevational gradients throughout the region based on ecological characteristics and conservation status. We identified a total of 2375 potential climate adaptation corridors linking lowland protected areas ($\geq$ 5 km$^2$) to high-elevation, contiguous forest patches ($\geq$ 5 km$^2$). Of these, 160 were deemed high-priority corridors owing to their greater protection, intact vegetation, wide

**Table 2. Ecological characteristics and conservation status of potential climate adaptation corridors and associated end and start nodes by corridor priority index level\*.**

| | Corridor variables | | | | | | |
|---|---|---|---|---|---|---|---|
| Level | Length (km) | Elevation range (m) | Forest biomass (Mt C) | Protection (%) | Overlapping protected areas | Overlapping KBAs | Number of corridors |
| High | 12, 169, 1158 | 1431, 2185, 3942 | 30, 64, 117 | 1, 46, 100 | 1, 9, 28 | 0, 6, 13 | 160 |
| Medium | 6, 123, 896 | 1241, 1868, 2975 | 17, 61, 114 | 1, 37, 100 | 1, 4, 25 | 0, 2, 10 | 925 |
| Low | 17, 194, 659 | 1323, 1829, 2565 | 24, 60, 98 | 1, 38, 87 | 1, 6, 22 | 0, 3, 7 | 1290 |
| | End node variables | | | | | | |
| Level | Area (km$^2$) | Elevation range (m) | Protection (%) | | | | |
| High | 5, 122, 5938 | 331, 1277, 3184 | 0, 100, 100 | | | | |
| Medium | 5, 11, 5938 | 143, 682, 3184 | 0, 5, 100 | | | | |
| Low | 5, 11, 754 | 232, 607, 1763 | 0, 63, 100 | | | | |
| | Start node variables | | | | | | |
| Level | Area (km$^2$) | Elevation range (m) | Protection (%)^ | | | | |
| High | 6, 122, 7226 | 2, 31, 607 | 100, 100, 100 | | | | |
| Medium | 5, 11, 7226 | 4, 171, 627 | 100, 100, 100 | | | | |
| Low | 5, 11, 4002 | 6, 137, 627 | 100, 100, 100 | | | | |

\* in columns 2–8, first, second, and third numbers are minimum, median and maximum values, respectively

^ start nodes were selected as protected areas

elevational gradients, and overlap with Key Biodiversity Areas. High-priority corridors also connected larger lowland protected areas to potential high-elevation climate change refugia that were generally larger, more protected, and spanned wider elevational gradients. However, these high-priority corridors were long (502 km on average) and often spanned at least 2 countries. Additionally, high-priority corridors were on average only 52% protected, with only 12 high-priority corridors fully protected. Therefore, future access to potential high-elevation refugia will likely be difficult for many species, even under the best of current landscape conditions.

Our results also indicate that many potential high-elevation refugia in Central America are currently small, fragmented, and poorly protected. Although we designed our corridor prioritization to identify corridors that generally terminated in larger, more protected, and more topographically heterogeneous end nodes, such locations are currently rare. Median individual end node size and protection in our analysis were just 11 km$^2$ and 4%, respectively. Given the small size of these sites, competition for local resources could limit successful range shifts even among species that can access these areas. Furthermore, the lack of protection of potential high-elevation refugia indicates that these sites could soon be degraded or lost. These results highlight the need to expand the current protected area network and increase conservation efforts to restore and enhance ecological connectivity among potential high-elevation refugia across Central America. A recent analysis in South America also demonstrated that most potential refugia across the region are currently unprotected [60]. Taken together, these findings depict a bleak picture for biodiversity under climate change throughout Latin America if additional steps to expand protected areas, increase their effectiveness for biodiversity outcomes, and ramp up conservation actions for climate change mitigation are not undertaken.

## Toward implementation of a regional climate adaptation corridor network in Central America

Some corridor studies have been previously criticized for being too coarse or conceptual to translate readily to on-the-ground conservation action, providing few tangible, practical

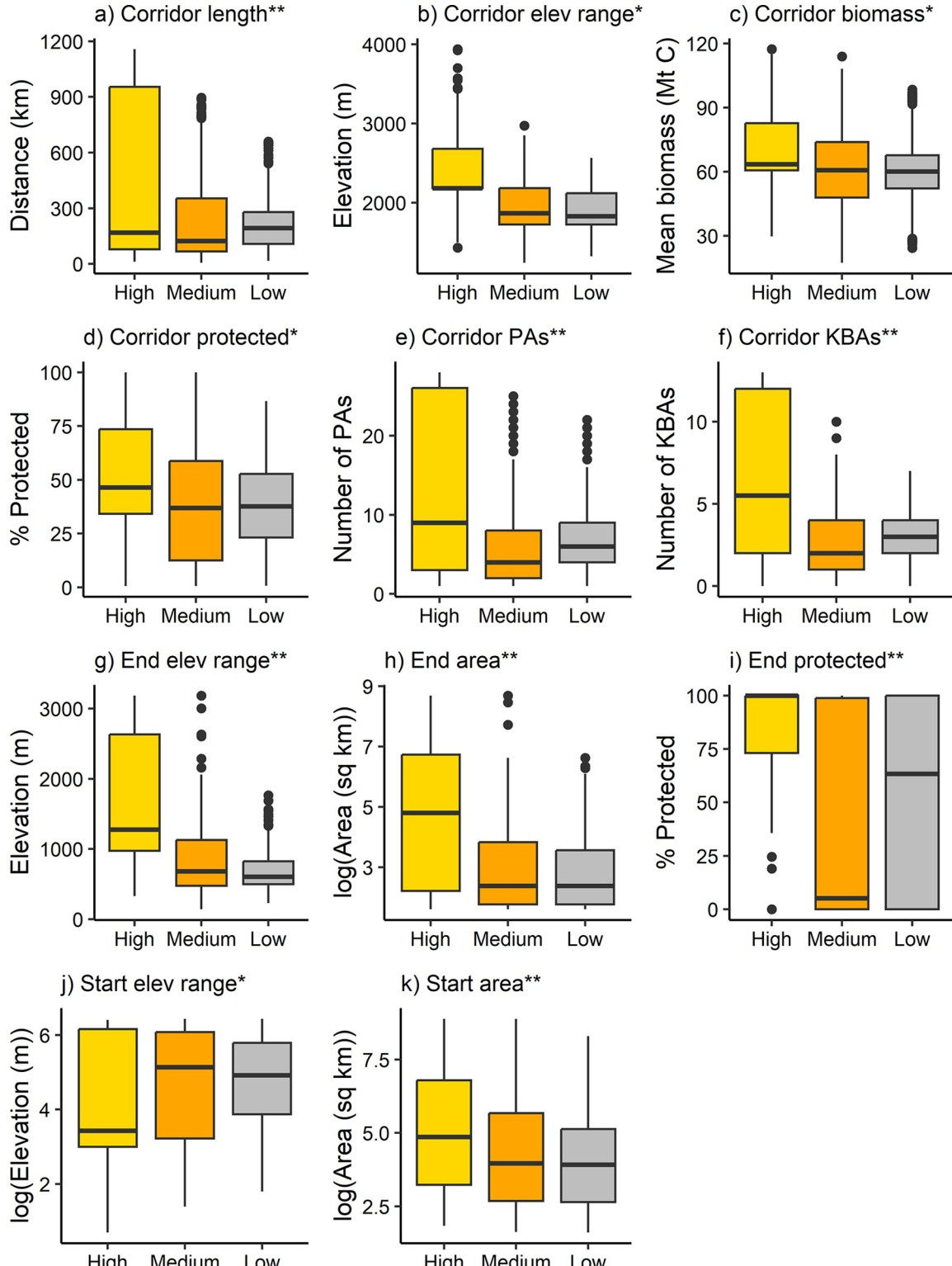

**Fig 5. Ecological characteristics and conservation status of potential climate adaptation corridors and end and start nodes.**
PA = protected area, KBA = Key Biodiversity Area. ** = significant differences among all group pairs, * = significant differences
between high and medium and high and low group pairs only (all based on Dunn's tests). Log transformations were for visual purposes
only.

strategies to implementing corridors [26, 61]. We helped address this by prioritizing potential climate adaptation corridors based on the ecological characteristics and the conservation status of individual corridors and their associated lowland protected area starting points and potential high-elevation refugia destinations. In doing so, we identified several 'high-priority landscapes' that offer strong potential as critical components of a regional climate adaptation corridor network in Central America due to their relatively high concentrations of high-priority corridors (S3 Fig in S1 Appendix). These landscapes can represent starting points for land managers and conservation practitioners aiming to implement climate adaptation corridors by highlighting where coordinated research, investment, conservation, or restoration activities could simultaneously benefit biodiversity resilience under climate change for multiple lowland protected areas. It is important to note, however, that our analysis prioritized potential corridors based on existing ecological landscape conditions, resulting in an uneven distribution of high-priority landscapes throughout Central America. Many other landscapes, such as portions of El Salvador or western Nicaragua, should still serve important roles in regional connectivity, particularly to leverage inclusivity and participation across all states and countries.

There are many factors to consider when moving from the regional scale to that of individual landscapes. The fact that even high-priority corridors are typically long and span multiple countries underscores the importance of international coordination for realizing a regional climate adaptation corridor network. Specifically, clear and coordinated objectives, management, and monitoring plans are necessary for all corridor initiatives, but particularly when envisioning a regional corridor network [5]. Weak, inconsistent, or overall lack of these things contributed substantially to the failures of the MBC [23]. Whereas our analyses applied global- or regional-scale geospatial datasets available for the entire study area, landscape-scale analyses would also benefit from more detailed integration of biodiversity data (e.g., species distributions, wildlife tracking, or camera trap data), forest characteristics, land-use types and agricultural practices, or barriers to fine-scale connectivity [62–64]. Other considerations could include loss of water resources due to climate or land use change (e.g., [65]), artificial light (e.g., [66]), or urban infrastructure such as buildings or roads (e.g., [67]). Such information could be used to determine necessary widths of individual corridors or be incorporated into corridor management and monitoring plans to track outcomes across different corridor projects throughout the region. Socio-economic costs and benefits of conservation and restoration activities associated with implementing corridors must also be considered to ensure stakeholder support and project durability [5]. Corridor projects may also be more likely to succeed when recognizing climate adaptive co-benefits for biodiversity and human well-being, including forest restoration and extreme temperature mitigation [68–70].

Successful implementation of climate adaptation corridors will require not only government and international support, but also a strong network of on-the-ground, site-based implementers that have been working in and understand the unique local realities that each landscape faces. Although local-scale climate adaptation corridor initiatives are largely in their infancy, illustrative examples are beginning to appear. In Costa Rica, efforts are ongoing to link lowland protected areas in the Osa Peninsula to the high-elevation La Amistad International Park (known as the ´AmistOsa´ landscape), and that includes the AmistOsa Biological Corridor designated by the Government of Costa Rica in 2017 [71]. The lowland protected areas have become isolated through deforestation and rapid encroachment by large and small-scale agriculture over the last 30 years, resulting in a mosaic of fragmented old growth and secondary forest, sun coffee farms, cattle pastures, and industrial-scale pineapple, teak, and oil palm plantations. Using the existing AmistOsa landscape as an example of on-the-ground initiatives to rebuild ecological connectivity along elevational gradients includes developing a habitat restoration network with over 200 local stakeholders to maintain and increase forest

cover within agricultural landscapes, supporting sustainable agricultural practices through economic incentives, restoring wetlands, reintroducing key focal species, and promoting civil responsibility among local communities to understand, advocate for, and protect their natural resources [72]. Whereas there is certainly no one-size-fits-all approach for facilitating corridor implementation, a regional climate adaptation corridor network initiative across Central America will need not only government and international support, but a strong network of local implementers familiar with local challenges and context.

## Conclusion

Our regional-scale analysis of potential climate adaptation corridors identified 2375 potential corridors throughout Central America that link lowland protected areas with contiguous, high-elevation forests. Although these high-elevation forests may represent potential climate change refugia owing to their relatively wide elevational gradients and potential climate buffering capacity, they are currently fragmented and poorly protected. Our climate adaptation corridor priority index considered the ecological characteristics and protected status of potential corridors and their associated start and end points, resulting in 160 high-priority corridors that are generally well protected and forested, span wide elevational gradients, overlap with Key Biodiversity Areas, and provide connections to habitats that are also generally more protected, span wide elevational gradients, and larger. However, the general lack of protection for potential high-elevation refugia throughout the region underscores the importance of not just promoting connectivity across elevational gradients, but also the urgency of additional conservation in the mountainous landscapes that need to be connected. Finally, it is critical that connectivity research in Central America does not stop at the regional scale. Whereas our analysis can serve as an umbrella guide for regional connectivity by identifying high-priority landscapes for future, finer-scale research, conservation, restoration, and investment, the implementation of climate adaptation corridors throughout Central America will require a coordinated suite of local- to landscape-scale efforts that collectively comprise the regional corridor network necessary to promote biodiversity resilience under climate change. Rebuilding ecological connectivity across elevational gradients can be a fundamental to tackling the biodiversity and climate crises together, a critical tool for achieving the 2030 Agenda for Sustainable Development, and a crucial nature-based solution for climate change mitigation, resilience, and adaptation that can also play a fundamental role in achieving key global conservation targets such as 30% protection by 2030 [20].

## Supporting information

**S1 Appendix. Mapping climate adaptation corridors for biodiversity—A regional-scale case study in Central America.** Supporting document containing supplementary figures and tables.
(DOCX)

## Acknowledgments

We thank an anonymous reviewer for comments and suggestions that improved this manuscript.

## Author Contributions

**Conceptualization:** Ian M. McCullough, Christopher Beirne, Carolina Soto-Navarro, Andrew Whitworth.

**Data curation:** Ian M. McCullough, Christopher Beirne.

**Formal analysis:** Ian M. McCullough, Christopher Beirne.

**Funding acquisition:** Andrew Whitworth.

**Investigation:** Ian M. McCullough, Christopher Beirne, Andrew Whitworth.

**Methodology:** Ian M. McCullough, Christopher Beirne, Carolina Soto-Navarro, Andrew Whitworth.

**Project administration:** Ian M. McCullough, Christopher Beirne, Carolina Soto-Navarro, Andrew Whitworth.

**Resources:** Ian M. McCullough, Christopher Beirne, Andrew Whitworth.

**Software:** Ian M. McCullough, Christopher Beirne.

**Supervision:** Ian M. McCullough, Christopher Beirne, Andrew Whitworth.

**Validation:** Ian M. McCullough, Christopher Beirne.

**Visualization:** Ian M. McCullough, Christopher Beirne.

**Writing – original draft:** Ian M. McCullough, Christopher Beirne, Carolina Soto-Navarro, Andrew Whitworth.

**Writing – review & editing:** Ian M. McCullough, Christopher Beirne, Carolina Soto-Navarro, Andrew Whitworth.

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
