## [Decision Letter · Decision Letter 0]

7 May 2024

PONE-D-24-09375Mapping climate adaptation corridors for biodiversity - A regional-scale case study in Central AmericaPLOS ONE

Dear Dr. McCullough,

Thank you for submitting your manuscript to PLOS ONE. After careful consideration, we feel that it has merit but does not fully meet PLOS ONE’s publication criteria as it currently stands. Therefore, we invite you to submit a revised version of the manuscript that addresses the points raised during the review process.

We look forward to receiving your revised manuscript.

Kind regards,

Lyi Mingyang, Ph.D.

Academic Editor

PLOS ONE

“This research was supported by the International Conservation Fund of Canada, the Bobolink Foundation, the BAND Foundation, and the Gordon and Betty Moore Foundation.”

“This research was supported by the International Conservation Fund of Canada, the Bobolink Foundation, the BAND Foundation, and the Gordon and Betty Moore Foundation.

BAND Foundation: https://bandfdn.org/

Bobolink Foundation: https://www.bobolinkfoundation.org/

International Conservation Fund of Canada: grant #2021-63; https://icfcanada.org/

Gordon and Betty Moore Foundation: grant #9075; https://www.moore.org/ 

All of these were gifts to Friends of the Osa (Osa Conservation) rather than to individual authors of this study. The BAND Foundation and Bobolink Foundation are small, family-run organizations that do not issue grant numbers per se. Sponsors or funders played no role in the study design, data collection and analysis, decision to publish, or preparation of the manuscript.”

4. We note that Figures 1, 4, S1 and S3 in your submission contain [map/satellite] images which may be copyrighted. All PLOS content is published under the Creative Commons Attribution License (CC BY 4.0), which means that the manuscript, images, and Supporting Information files will be freely available online, and any third party is permitted to access, download, copy, distribute, and use these materials in any way, even commercially, with proper attribution. For these reasons, we cannot publish previously copyrighted maps or satellite images created using proprietary data, such as Google software (Google Maps, Street View, and Earth). For more information, see our copyright guidelines: http://journals.plos.org/plosone/s/licenses-and-copyright.

1. You may seek permission from the original copyright holder of Figures 1, 4, S1 and S3 to publish the content specifically under the CC BY 4.0 license. 

Additional Editor Comments:

Dear authors,

thank you for the submission of your interesting manuscript to PlosOne.

The reviewer finds the content rather compelling while indicating critical points to address. The reviewer provided very useful suggestions to improve the overall clarity of your study as well as the quality of your analysis. The suggestions of the reviewer look feasible to me, and I believe you will be able to address them. Thus, please take care to do a full revision of your manuscript according to all reviewer's comments. Improvements based on reviewers’ comments will be crucial for acceptance.

Best regards,

LM

Reviewers' comments:

Reviewer's Responses to Questions

**Comments to the Author**

1. Is the manuscript technically sound, and do the data support the conclusions?

Reviewer #1: Yes

2. Has the statistical analysis been performed appropriately and rigorously? 

Reviewer #1: Yes

3. Have the authors made all data underlying the findings in their manuscript fully available?

Reviewer #1: Yes

4. Is the manuscript presented in an intelligible fashion and written in standard English?

Reviewer #1: Yes

5. Review Comments to the Author

Reviewer #1: In the paper “Mapping climate adaptation corridors for biodiversity - A regional scale case study in Central America” the authors identified 2375 potential corridors throughout Central America that link lowland protected areas (≤ 500 m) with intact, high-elevation forests (≥ 1500 m) that represent potential climate change refugia. We also developed a corridor priority index based on the ecological characteristics and protected status of potential corridors and their associated start and end points. This manuscript is well organized, and the drawn conclusions are coherent with the obtained results. I hope to provide very useful suggestions to improve the overall clarity of your study as well as the quality of your analysis. I think that my suggestions look feasible to you, and I believe you will be able to address them. Thus, please take care to do a full revision of your manuscript according to all my comments. Improvements based on my comments will be crucial for acceptance. I have some concerns and suggestions for each aspect of the manuscript. Please see below.

Lines 21 – 36: Please give more emphasis to your results.

Lines 37 - 123: The introduction is technically sound and the claims are convincing. However I think that some references should be updated. Please, note that the hypothesis and the predictions are unclear, you need to well explain them.

Lines 45 - 46: I think that you should add these important references to support your sentence: “The basis of this strategy has arisen from a body of science related to paleogeography, paleoecology, and past, present, and future climate and species distribution modeling”. I would like to suggest:

Bosso, L., et al. (2024). Integrating citizen science and spatial ecology to inform management and conservation of the Italian seahorses. Ecological Informatics, 79, 102402.

Gracanin, A., & Mikac, K. M. (2023). Evaluating modelled wildlife corridors for the movement of multiple arboreal species in a fragmented landscape. Landscape Ecology, 38(5), 1321-1337

Lines 113 – 123: Please, explain in detail your hypothesis and predictions. You need to expand this section if you would want to express exactly what you want to do.

Lines 125 – 259: Materials and methods: In general, the methods are appropriate and the study seems well conducted, although some details deserve a bit more attention i.e., especially about the methodology and the data. All the scripts/codes used in this paper must be added in the supplementary materials. Please, provide also all the link to source where you downloaded the data.

Lines 201 – 203: I think that you should add these important references to support your sentence: “Like many previous regional- to continental-scale climate connectivity studies, we did not adjust conductance values for topography as our focus was long-term range shifts rather than short-term movements”. I would like to suggest:

Kabir, M., et al., (2017). Habitat suitability and movement corridors of grey wolf (Canis lupus) in Northern Pakistan. PloS one, 12(11), e0187027.

Bongi, P., et al., (2023). Coexistence in ecological corridors: understanding tolerance of wolves in the Northwestern Apennines, Italy. Human Dimensions of Wildlife, 28(1), 53-69.

Lines 260 – 374: Well written! The figures and the tables are all informative and necessary, but not redundant, ensuring the correct comprehension of the manuscript.

Lines 376 – 491: The paper discussed appropriately the context and the theme, although there is important literature not cited by the authors. I think that the authors should discuss their results also comparing them with those already published on other species/genus/family. In fact your paper discusses findings in relation to some of the work in the field but ignores other important work that I think should be added in your discussion.

Lines 427 – 428 : I think that you should add these important references to support your sentence: “ There are many factors to consider when moving from the regional scale to that of individual landscapes.”. I would like to suggest:

Fraissinet, M., et al., (2023). Responses of avian assemblages to spatiotemporal landscape dynamics in urban ecosystems. Landscape Ecology, 38(1), 293-305.

Wojkowski, J., et al., (2023). Are we losing water storage capacity mostly due to climate change–Analysis of the landscape hydric potential in selected catchments in East-Central Europe. Ecological Indicators, 154, 110913.

Lines 433 – 437 : I think that you should add these important references to support your sentence: “Whereas our analyses applied global- or regional-scale geospatial datasets available for the entire study area, landscape-scale analyses would also benefit from more detailed integration of biodiversity data (e.g., species distributions, wildlife tracking, or camera trap data), forest characteristics, land-use types and agricultural practices, or barriers to fine-scale connectivity (e.g., roads).”. I would like to suggest:

Salinas-Ramos, V. B., et al., (2021). Artificial illumination influences niche segregation in bats. Environmental Pollution, 284, 117187.

Ehlers Smith, Y. C., et al., (2018). Forest habitats in a mixed urban-agriculture mosaic landscape: patterns of mammal occupancy. Landscape Ecology, 33, 59-76.

6. PLOS authors have the option to publish the peer review history of their article (what does this mean?). If published, this will include your full peer review and any attached files.

Reviewer #1: No

---

## [Author Response · Author response to Decision Letter 0]

10 May 2024

Response to Reviewers

Reviewers' comments:

Reviewer #1: In the paper “Mapping climate adaptation corridors for biodiversity - A regional scale case study in Central America” the authors identified 2375 potential corridors throughout Central America that link lowland protected areas (≤ 500 m) with intact, high-elevation forests (≥ 1500 m) that represent potential climate change refugia. We also developed a corridor priority index based on the ecological characteristics and protected status of potential corridors and their associated start and end points. This manuscript is well organized, and the drawn conclusions are coherent with the obtained results. I hope to provide very useful suggestions to improve the overall clarity of your study as well as the quality of your analysis. I think that my suggestions look feasible to you, and I believe you will be able to address them. Thus, please take care to do a full revision of your manuscript according to all my comments. Improvements based on my comments will be crucial for acceptance. I have some concerns and suggestions for each aspect of the manuscript. Please see below.

Response: Thank you for your thorough and thoughtful review of the manuscript. Your comments and suggestions have helped us improve the paper considerably. We appreciate the time and effort you put into this. Please note that line references in our responses below refer to the clean (i.e., not track-changed) revised manuscript.

Lines 21 – 36: Please give more emphasis to your results.

Response: We added some more specific information about corridor protection and characteristics of high-priority corridors (revised lines 26-29, 31-37). We also added specific information to support the assertion that highland target areas are unprotected and small but span wider elevational gradients compared to lowland protected areas (revised lines 37-41). These additions got the abstract up to 295 words (abstract word limit is 300 words). We deemed that these revisions struck a balance between providing specific statistics and a general summary of the paper’s key findings.

Lines 37 - 123: The introduction is technically sound and the claims are convincing. However I think that some references should be updated. Please, note that the hypothesis and the predictions are unclear, you need to well explain them.

Response: Thank you for this comment. We added a specific hypothesis in the last paragraph of the introduction (revised lines 117-120).

Lines 45 - 46: I think that you should add these important references to support your sentence: “The basis of this strategy has arisen from a body of science related to paleogeography, paleoecology, and past, present, and future climate and species distribution modeling”. I would like to suggest:

Bosso, L., et al. (2024). Integrating citizen science and spatial ecology to inform management and conservation of the Italian seahorses. Ecological Informatics, 79, 102402.

Gracanin, A., & Mikac, K. M. (2023). Evaluating modelled wildlife corridors for the movement of multiple arboreal species in a fragmented landscape. Landscape Ecology, 38(5), 1321-1337

Response: Thank you for these interesting references. We added them to support the sentence you referenced (revised lines 51-53).

Lines 113 – 123: Please, explain in detail your hypothesis and predictions. You need to expand this section if you would want to express exactly what you want to do.

Response: We added a specific hypothesis in the last paragraph of the introduction (revised lines 117-120). This includes a new phrase about the importance of identifying priority corridors. The rest of this paragraph describes the applied conservation importance of this study. We feel that as a whole, this revised paragraph now succinctly summarizes the purpose and applied value of this study. Thank you for your suggestions.

Lines 125 – 259: Materials and methods: In general, the methods are appropriate and the study seems well conducted, although some details deserve a bit more attention i.e., especially about the methodology and the data. All the scripts/codes used in this paper must be added in the supplementary materials. Please, provide also all the link to source where you downloaded the data.

Response: In compliance with the journal’s policy, we created a preliminary, permanent online repository that contains data and R code. We cited this repository in the methods. The journal recommends that we use a permanent online data repository rather than supplementary materials attached to the manuscript.

McCullough I, Beirne C. Central America climate adaptation corridor analysis (beta). Zenodo; 2024. doi: 10.5281/zenodo.10794501.

For our revision, we have updated this repository and cited it in the methods (revised lines 155-156):

McCullough I, Beirne C. Central America climate adaptation corridor analysis (1.0). Zenodo; 2024. doi: 10.5281/zenodo.11150568.

Lines 201 – 203: I think that you should add these important references to support your sentence: “Like many previous regional- to continental-scale climate connectivity studies, we did not adjust conductance values for topography as our focus was long-term range shifts rather than short-term movements”. I would like to suggest:

Kabir, M., et al., (2017). Habitat suitability and movement corridors of grey wolf (Canis lupus) in Northern Pakistan. PloS one, 12(11), e0187027.

Bongi, P., et al., (2023). Coexistence in ecological corridors: understanding tolerance of wolves in the Northwestern Apennines, Italy. Human Dimensions of Wildlife, 28(1), 53-69.

Response: Although we appreciate these suggestions, this sentence was intended to make the specific point about not adjusting conductance values for topography in climate connectivity modeling. The three studies we cited represent successful, published examples of studies that did not make such adjustments. We did not cite Kabir et al. (2017) because they used a terrain variable (“vector ruggedness”) in the habitat suitability model that was the basis for their conductance surface used in connectivity modeling. We also did not cite Bongi et al. (2023) because it is not a study about modeling connectivity per se. Although it is a quite interesting article, it is a stakeholder survey study that is not directly relevant to the point we are making here. Nonetheless, thank you for these interesting suggestions.

Lines 260 – 374: Well written! The figures and the tables are all informative and necessary, but not redundant, ensuring the correct comprehension of the manuscript.

Response: We appreciate this feedback.

Lines 376 – 491: The paper discussed appropriately the context and the theme, although there is important literature not cited by the authors. I think that the authors should discuss their results also comparing them with those already published on other species/genus/family. In fact your paper discusses findings in relation to some of the work in the field but ignores other important work that I think should be added in your discussion.

Response: We interpreted this comment as a general comment about the discussion. Our responses below to the other, more specific comments about the discussion cover the new references that we added. Thank you for these suggestions.

Lines 427 – 428 : I think that you should add these important references to support your sentence: “ There are many factors to consider when moving from the regional scale to that of individual landscapes.”. I would like to suggest:

Fraissinet, M., et al., (2023). Responses of avian assemblages to spatiotemporal landscape dynamics in urban ecosystems. Landscape Ecology, 38(1), 293-305.

Wojkowski, J., et al., (2023). Are we losing water storage capacity mostly due to climate change–Analysis of the landscape hydric potential in selected catchments in East-Central Europe. Ecological Indicators, 154, 110913.

Response: We added the Wojkowski et al. and Fraissinet et al. references to support the sentences about factors to consider in finer-scale connectivity studies (revised lines 442-448). In response to the comment above about providing references for other taxa, in addition to this study about birds, we also added a reference to a multi-species mammal study and an amphibian study:

Brodie JF, Giordano AJ, Dickson B, Hebblewhite M, Bernard H, Mohd‐Azlan J, Anderson J, Ambu L. Evaluating multispecies landscape connectivity in a threatened tropical mammal community. Conservation Biology. 2015 Feb;29(1):122-32. 

Matos C, Petrovan SO, Wheeler PM, Ward AI. Landscape connectivity and spatial prioritization in an urbanising world: A network analysis approach for a threatened amphibian. Biological Conservation. 2019 Sep 1;237:238-47.

Lines 433 – 437 : I think that you should add these important references to support your sentence: “Whereas our analyses applied global- or regional-scale geospatial datasets available for the entire study area, landscape-scale analyses would also benefit from more detailed integration of biodiversity data (e.g., species distributions, wildlife tracking, or camera trap data), forest characteristics, land-use types and agricultural practices, or barriers to fine-scale connectivity (e.g., roads).”. I would like to suggest:

Salinas-Ramos, V. B., et al., (2021). Artificial illumination influences niche segregation in bats. Environmental Pollution, 284, 117187.

Ehlers Smith, Y. C., et al., (2018). Forest habitats in a mixed urban-agriculture mosaic landscape: patterns of mammal occupancy. Landscape Ecology, 33, 59-76.

Response: Thank you for these suggestions. We added these references to revised lines 446-448 about considerations for fine-scale landscape connectivity research. We agree that it is important to consider artificial light and urban infrastructure when mapping landscape-scale connectivity, particularly in human-dominated landscapes.

---

## [Editor Report · Decision Letter 1]

20 May 2024

Mapping climate adaptation corridors for biodiversity - A regional-scale case study in Central America

PONE-D-24-09375R1

Dear Dr. McCullough,

We’re pleased to inform you that your manuscript has been judged scientifically suitable for publication and will be formally accepted for publication once it meets all outstanding technical requirements.

Kind regards,

Lyi Mingyang, Ph.D.

Academic Editor

PLOS ONE

Additional Editor Comments (optional):

Well done! The authors responded effectively to all of the reviewer comments. Now the manuscript is ready for publication in PlosOne.

LM
---

## [Editor Report · Acceptance letter]

22 May 2024

PONE-D-24-09375R1 

PLOS ONE

Dear Dr. McCullough, 

I'm pleased to inform you that your manuscript has been deemed suitable for publication in PLOS ONE. Congratulations! Your manuscript is now being handed over to our production team.

Kind regards, 

on behalf of

Professor Lyi Mingyang 

Academic Editor

PLOS ONE